# Health Information Technology Usability Evaluation Scale (Health-ITUES) and User-Experience Questionnaire (UEQ) for 3D Intraoperative Cognitive Navigation (ICON3D^TM^) System for Urological Procedures

**DOI:** 10.3390/medicina59030624

**Published:** 2023-03-21

**Authors:** Enrico Checcucci, Federico Piramide, Sabrina De Cillis, Gabriele Volpi, Alberto Piana, Paolo Verri, Andrea Bellin, Michele Di Dio, Cristian Fiori, Francesco Porpiglia, Daniele Amparore

**Affiliations:** 1Department of Surgery, Candiolo Cancer Institute, FPO-IRCCS, 10060 Candiolo, Italy; 2Division of Urology, Department of Oncology, School of Medicine, San Luigi Hospital, University of Turin, 10043 Orbassano, Italy; 3Division of Urology, Department of Surgery, Annunziata Hospital, 10060 Cosenza, Italy

**Keywords:** augmented reality, 3D models, laparoscopy, 3D guided surgery

## Abstract

*Backgound and objectives:* In recent years, the adoption of 3D models for surgical planning and intraoperative guidance has gained a wide diffusion. The aim of this study was to evaluate the surgeons’ perception and usability of ICON3D^TM^ platform for robotic and laparoscopic urological surgical procedures. *Materials and Methods:* During the 10th edition of the Techno-Urology Meeting, surgeons and attendees had the opportunity to test the new ICON3D^TM^ platform. The capability of the user to manipulate the model with hands/mouse, the software usability, the quality of the 3D model’s reproduction, and the quality of its use during the surgery were evaluated with the Health Information Technology Usability Evaluation Scale (Health-ITUES) and the User-Experience Questionnaire (UEQ). *Results:* Fifty-three participants responded to the questionnaires. Based on the answers to the Health-ITUES questionnaire, ICON3D^TM^ resulted to have a positive additional value in presurgical/surgical planning with 43.4% and 39.6% of responders that rated 4 (agree) and 5 (strongly agree), respectively. Regarding the UEQ questionnaire, both mouse and infrared hand-tracking system resulted to be easy to use for 99% of the responders, while the software resulted to be easy to use for 93.4% of the responders. *Conclusions:* In conclusion, ICON3D^TM^ has been widely appreciated by urologists thanks to its various applications, from preoperative planning to its support for intraoperative decision-making in both robot-assisted and laparoscopic settings.

## 1. Introduction

Over the last years, the advent of robotics has revolutionized the approach to minimally invasive surgery, in particular in urology, and has gained an ever-wider popularity among surgeons, especially for complex surgical procedures such as radical prostatectomy [1] and partial nephrectomy [2]. New technologies have allowed us to pursue even greater results, introducing the concept of surgical navigation [3,4]. In fact, the implementation of 3D models’ use during robot-assisted surgical procedures seems to have furtherly improved functional and oncological results of such procedures, thanks to a better visualization of both healthy and pathological structures [5,6]. 

For renal cancer surgery, the 3D models allow the enhancement of the arterial vasculature of the kidney for perioperative surgical planning during robotic partial nephrectomy, especially for the treatment of high-complexity renal masses. Thanks to the use of these models, a significantly lower rate of patients underwent global ischaemia in comparison to the 2D group [2]. Moreover, 3D virtual models gave a better understanding of tumour location, endophytic rate, and relationship with vessels before surgery, leading to an increased indication for nephron-sparing surgery. If it is evident in a cognitive setting, even more importance of the 3D models was proven in an augmented reality setting in robotics. In fact, for totally endohitic renal masses the overlapped images can allow to identify the lesion in real-time and to guide the surgeon during the resection phase. Moreover, the intraparenchymal structures such as vessels and calyxes can be visualized and sutured. 

Focusing on prostate cancer, the 3D models obtained from MRI images permit us to visualize the lesion location and its relationship with the prostate capsule. Especially in an augmented reality setting thanks to the 3D images is possible to identify the extracapsular extension of the lesion and to modulate selectively the nerve-sparing phase or to perform 3D guided selective biopsies.

However, today, pure laparoscopy still represents the most common surgical approach in urology, meeting continuous attempts of renewal and technological updating with the introduction of new laparoscopic instruments and new cameras, aiming to obtain ever more detailed images of the intraoperative field [7,8,9]. However, the application of real-time use of 3D models during laparoscopic surgery remains anecdotal. Some pioneering experiences have been published regarding the use of 3D models also in this setting [10], but an extensive adoption and application of this technology has been limited due to the difficult interaction between the scrubbed surgeons and the surgical navigation tools. With the aim to make the 3D technology usable both for robotics and laparoscopy, thanks to the collaboration with Medics3D’s (https://www.medics3d.com, accessed on 1 March 2022) bioengineers we have developed a specific platform for 3D models management and navigation, named ICON3D^TM^ (Intraoperative Cognitive Navigation), which ensures online cloud access to the 3D models and, thanks to a dedicated rack, allows surgeons to exploit independently the virtual reconstructions from preoperative planning to intraoperative cognitive navigation or augmented reality surgical procedures. The aim of this study was to evaluate the surgeons’ perceptions and usability of ICON3D^TM^ platform for robotic and laparoscopic urological surgical procedures.

## 2. Materials and Methods

### 2.1. Study Population

In the present study, all the participants of the 10th Techno-Urology Meeting, which took place at San Luigi Gonzaga Hospital, Orbassano, Turin-Italy, and Candiolo Cancer Institute, Candiolo-Italy, in April 2022 (http://www.technourologymeeting.com, accessed on 1 February 2022) (both surgeons and attendees) had the possibility to have the ICON3D^TM^ experience, in a dedicated area.

Close to the auditorium, a specific space was set up. The attendees had the possibility to download and navigate the 3D models using the ICON3D^TM^ platform. Moreover, the ICON3D^TM^ rack was available in that area, simulating a laparoscopic or robotic setting. Thanks to a prerecorded intraoperative video, the participants could enjoy the model during the prerecorded surgery in a cognitive or augmented reality (AR) manner.

The attendees after their experience with the platform were asked to fill a dedicated questionnaire; those who participated in the study were classified according to their level of expertise as follows: residents [Re], young urologists [YU], senior urologists [SU] (more than 10 years as urologists); and on the basis of their surgical skills (open, laparoscopic, robotic). Written consent and IRB approval were not required for the present study.

### 2.2. 3D Models Reconstruction

Since it provides a thorough picture of the patient’s anatomy, radiological imaging, such as CT or MRI, is a crucial stage in the diagnosis and treatment planning of the majority of urological diseases. However, it can be challenging for novice urologists in particular to have a precise understanding of the anatomical structures. In fact, a surgeon must follow a learning curve that takes time to walk in order to undertake an effective “building in mind” process [11]. When compared to 2D CT/MRI images, 3D reconstructions, as intuitive as they can be, offer information that is easier to access: nearby organ proportions and relationships are more understandable, and the pathology itself (whether malignant or benign) can be displayed and visualized in a different way.

Bidimensional images are the beginning point for creating 3D models. All common DICOM viewer software comes with the option to automatically draw three-dimensional reconstructions, however, the quality is frequently mediocre. Despite their poor quality, these models can be helpful in overcoming the limitations of the 2D slices by providing additional information and details through a spatial depiction of the organs and the characteristics of the disease. The introduction of a new player—the specialized bioengineer—to the team was necessary in order to create the more realistic models that surgeons often require. This new team member can offer a service designed to produce models that are more accurate in terms of detailing and anatomy. The capacity of the doctors and engineers to communicate effectively is essential to the process’ success. In order to develop an accurate computer project, it is critical that the engineers comprehend the needs of the surgeon and vice versa. Practically speaking, the collection of 2D photographs of the patient is the first step in the model’s realization.

When focusing on kidney cancer, CT scans (multi-slice is desirable) that can be easily exported in DICOM format provide the most valuable information. MRI pictures can also be employed.

The fidelity of the 3D reconstruction is increased by high-quality imaging, and a single slice’s thickness should not be more than 5 mm.

The object must first be inspected and researched in 2D using DICOM pictures displaying software, choosing the most helpful images (such as the arterial or late phase of a CT-scan) and altering certain parameters (such as image contrast and luminance) in accordance with the objectives of the project. Preprocessing phase is the name of this stage.

Following that, a volume rendering is produced. During this stage, the software automatically creates a preliminary 3D model utilizing data from the picture voxels. The fundamental volume unit, or voxel, is comparable to a pixel in a 2D system. This preliminary representation provides a general concept of the project and enables the engineer to pinpoint important problems.

After that, a procedure known as “segmentation” is carried out with the use of specialized software. Segmentation is the separation of pixels that are part of regions or objects of interest (ROIs/OOIs), chosen using an arbitrary similarity criterion (e.g., color). “Thresholding” is the procedure that makes it most simple to identify various ROIs and OOIs. The software may detect all the locations with the desired qualities by the engineer by using a certain range of a set parameter (such as gray scale). As a result, certain algorithms are created, and other locations and things are automatically removed. This is a crucial phase in the creation of 3D models since, in certain circumstances, the software is unable to accurately detect and represent the many elements, necessitating manual intervention. The engineer’s role is crucial at this point, and the knowledge they’ve gained enables them to customize the patient’s unique 3D model, just like a craftsman would with his finished product.

The project can then be exported and saved in.stl (Standard Triangulation Language) format, allowing the operator to make additional modifications to the rendering using specialized software after this step is complete. The virtual 3D model is finished at this point. The 3D model can then be downloaded and transferred to other electronic devices.

The prostatic glands, the prostatic lesion (even in situations where it was fully intra-prostatic), and the conformation of the neurovascular bundles were all replicated in 3D virtual models of the prostates starting from 2D MRI images for prostate cancer. An approach based on contours was used to segment the particular structures.

### 2.3. ICON3D^TM^ System

ICON3DTM is a SAMD (Software as a Medical Device) for the preoperative and intraoperative visualization and manipulation of 3D virtual anatomical models. In particular, the Hyper Accuracy Three-Dimensional (HA3D^®^) models, which are interactive and patients’ specific [11], can be exploited using the ICON3D^TM^ software.

The objective of this tool is to provide support for complex surgeries improving the user experience of the surgeons, allowing them to reproduce the pre-operative planning during surgery. In fact, thanks to the possibility to have a cloud storage and access to the models, surgeons can simulate the surgical planning (type of resection, clamping, suture) preoperatively; subsequently, the 3D models with the planned surgical strategy can be downloaded and exploited during the surgery. 

On the ICON3D^TM^ monitor, physicians can independently and interactively evaluate the type of resections and the relevant anatomical details, following the planned sequence of surgical gestures.

Concerning cognitive surgery, the intended use is as an auxiliary display. The surgeons can look at the ICON3D^TM^ monitor to increase their perception of the real patient’s anatomy, to better recognize the anatomical structures, and reproduce the pre-operative planning, previously orchestrated on MyMedics portal (www.mymedics3d.com). Furthermore, ICON3D^TM^ allows surgeons to visualize the 3D models overlapped to the endoscopic video streams, providing a manual landmark-based registration for AR solution.

The HA3D^®^ models, both in a cognitive or AR setting in robotics, can be exploited thanks to a dedicated mouse consolidated with the robotic console. On the other hand, models’ manipulation in laparoscopy occurs through a hand-tracking system in touchless mode with a codified hand gesture for laparoscopy: without touching any interface, preserving the sterility, the surgeon can have access to all the potentials of the software. 

Furthermore, in particular for laparoscopic procedures, compatible medical grade accessory hardware for the use of ICON3D^TM^ inside the operative room is available. The dedicated ICON3D^TM^ rack is composed of (1) a flexible adjustable arm that allows the user to maintain the best position for the ICON3D™ monitor for the entire surgical team (especially for laparoscopy); (2) the ICON3D™ Touchless Sensor Kit, designed to allow easy isolation from the sterile field using dedicated lids in combination with standard camera covers; (3) the ICON3D™ cart, equipped with high-end informatic components to ensure the best performance of the software even with intense use in AR mode. See Figure 1.

### 2.4. ICON3D^TM^ Evaluation Questionnaire 

All the participants to the congress who make use of the ICON3D™ experience were asked to fill the questionnaire reported in Appendix A. The questionnaire was structured in two different sections.

The first one (questions #1 to #24) is based on the Health Information Technology Usability Evaluation Model (Health-ITUEM) which assesses mobile health (mHealth) technologies [12]. The Health-ITUEM is an integrated model of multiple usability theories based on the concepts of usability from the Technology Acceptance Model [13] and the International Organization for Standardization (ISO) standard 9241-11 [14]. 

The Health Information Technology Usability Evaluation Scale (Health-ITUES) is a tool evaluating the task addressing different levels of expectation of support for the task by the health information technology (IT). Moreover, the Health-ITUES is customizable and can assess the study purposes without item addition, deletion, or modification. 

The second part of the questionnaire (questions from #25 to #41) was specifically designed in order to evaluate the usability of the software (user-experience questionnaire, UEQ) starting from the definition of the ICON3D^TM^ primary user functions, according to IEC 62366-2:2016 Table E.1 suggestions [15]. 

The questions explored the capability of the user to manipulate the model with hands/mouse, the software usability, the quality of the 3D model’s reproduction and the quality of its use during the surgery.

### 2.5. Statistical Analysis 

Descriptive statistics include frequencies and proportions. Differences between the answers of the three different groups of participants (Re, Yu, Su) were tested using Fisher’s exact test. Statistical significance was set as *p* < 0.05. Statistical analyses were all performed using the Jamovi software (version 2.3).

## 3. Results

### 3.1. Study Population

Fifty-three participants used the ICON3D^TM^ experience, of which 16 (30.8%) were female. A total of 31 (58.5%) were residents, whilst 16 (30.2%) and 6 (11.3%) were young urologists or seniors, respectively. The majority of them were revealed to be skilled in mini-invasive surgery: 21 (42.9%) in laparoscopy and 38 (77.6%) in robotics. See Table 1.

### 3.2. Health-ITUES Questionnaire

ICON3D^TM^ (Figure 2) resulted to have a positive adding value both for the patients and the surgeons in presurgical/surgical planning with 43.4% and 39.6% of responders who answered 4 (agree) and 5 (strongly agree), respectively to the questions 5, 6, and 7.

Focusing on “Perceived usefulness” of ICON3D^TM^ technology, it turned out to be a useful tool for surgical strategy (Q9 median score 5 [IQR 4–5], Q12 median score 4 [IQR 4–5]) and planning (Q11 median score 5 [IQR 4–5]), thanks to an optimal understanding of patient’s anatomy (Q8 median score 5 [IQR 4–5]) and the possibility to self-manage the model (Q16 median score 4 [IQR 4–5]).

Concerning the “Perceived ease of use” at Q17–20 33.0%, 38.7%, and 25.5% of the responders rated the ICON3D^TM^ with a score of 3, 4, or 5, respectively (*p* = 0.15). 

Evaluating the “User control” in terms of comprehension of error message (Q22) the median score was 4 (IQR 3–4); the capability to recover an error easily and quickly (Q23) was also rated 4 (IQR 3–4). The on-line help, on-screen messages, and other documentation (Q24) were evaluated with a median score of 4 (IQR 3–4).

### 3.3. UEQ Questionnaire

Both mouse or infrared hand-tracking systems turned out to be easy to use for 99% of the attendees (response at Q25, 26 with “absolutely yes” or “Yes, but it takes some time to become fully autonomous), and the hardest phase resulted to be the navigation in the menu and to break down the model to its part (Q28, 29). Furthermore, the software resulted to be easy to use for 93.4% of the responders (Q32, 33). The quality of the 3D models was optimally scored both in terms of colors and anatomical details as revealed by Q31 and Q34 with a low rate of risk of mistakes (Q36) or mistakes the anatomy (Q37) (Figure 3).

A total of 94.3% of the users were satisfied with their ability in software’s handling (Q38); furthermore, this software turned out not to be time consuming during the surgical procedure for 94.2% of the attendees and did not cause any stress in 90.4% of the physicians that exploited ICON3D^TM^ technology (Q40).

### 3.4. Evaluation of the Different Levels of Expertise

Considering the baseline characteristics of our population, no differences were found in terms of robotic (79 vs. 78 vs. 66% for Re, YU, and SU, respectively, *p* = 0.79) or laparoscopic skills (37.9% vs. 35.7 vs. 83% for residents, young urologists, and senior urologists, respectively, *p* = 0.10). 

By all three groups, the ICON3D^TM^ was judged to be a useful tool for both patients (*p* = 0.62) and surgeons (*p* = 0.84) (Q 5–6). Furthermore, 77.4%, 93%, and 66.6% (*p* = 0.73) of Re, YU, and SU, respectively, agreed (4) or strongly agreed (5) in considering the ICON3D^TM^ an important part of their presurgical/surgical planning (Q 7). 

The “perceived usefulness” confirmed these findings and the ICON3D^TM^ technology was deemed to be a useful tool for both surgical strategy (Q9 and Q12) and planning (Q11), without differences between the three groups (Q9 *p* = 0.84; Q11 *p* = 0.62; Q12 *p* = 0.12).

The better understanding of patients’ anatomy together with the self-management of the model were by far considered the biggest advantages of ICON3D^TM^(Q8 median Re, YU, and SU score were 5 [IQR 4–5], 4.5 [IQR 4–5], 4 [IQR 4–4.75] *p* = 0.38; and Q16 median Re, YU, and SU score 4 [IQR 3–4], 4 [IQR 4–5], 4.5 [IQR 3.25–5] *p* = 0.50). 

Furthermore, it was also considered easy to use by 48.4%, 56.3%, and 66.7% of Re, YU and SU, respectively (Q17 *p* = 0.43). In the same way, also the learning process was not deemed to be tough, with no differences between the three groups (Q20, *p* = 0.53). Finally, even stratifying the attendees by the level of surgical expertise, the UEQ did not reveal any major criticism for any group. The hardest phase (Q28) turned out to be the navigation in the menu and the breakdown of the model to its part for the majority of YU (46.7%), while moving/tilting the 3D model synchronously with the surgery was the hardest one for Re and SU (50% for both group). A total of 93.6%, 93.8%, and 100% of Re, YU and SU, respectively, rated the platform as not time-consuming (Q39, *p* = 0.85) and 83.3%, 100%, and 100% of them as not stressful (Q40 *p* = 0.27). See Figure 4 and Figure 5

## 4. Discussion

Herein we present for the first time the usability and users’ perception of the new specific platform for 3D models management and navigation, named ICON3D^TM^ (Intraoperative Cognitive Navigation). 

Regarding the Health-ITUES questionnaire, ICON3D^TM^ demonstrated to be helpful both for presurgical and surgical planning with 43.4% and 39.6% of the attendees who rated 4 (agree) and 5 (strongly agree), respectively. Moreover, it is worth mentioning that, despite the stratification on the basis of surgical expertise, the questions evaluating the usefulness of ICON3D^TM^ showed good results across the three groups (Re, YU, SU). In particular, the surgical anatomy was clearly understandable by all the groups. 

On the other hand, the software’s usability was evaluated with the UEQ questionnaire with excellent feedback by the attendees notwithstanding the different levels of expertise.

The development of this platform perfectly fits in the current technology-driven era of urologic surgery, where the 3D models have gained a wide diffusion during different steps of surgery starting from preoperative evaluation to patient counseling and intraoperative navigation [16]. If at first their application was particularly dedicated to oncologic surgery [6,17,18,19], more recently these technologies have attracted interest also for benign diseases [20].

An important consideration related to the clinical use of 3D models, in particular for the intraoperative setting, is that their implementation was at first dedicated to robotic surgery, especially in augmented reality procedures, thanks to the Tile-pro visualization mode [5,21,22]. 

However, as emerged in a recent survey, the laparoscopic approach still remains the most diffused across Europe [23], therefore the need to develop a visualization platform adapted also for this setting was felt. The ICON3D^TM^ with its dedicated rack allows us to visualize and navigate the 3D model during the intervention in a cognitive or augmented reality manner, and thanks to the Touchless Sensor Kit the scrubbed surgeon can handle the models autonomously.

Another mention should be reserved for the MyMedics cloud system that allows the sharing of CT/MRI images and for users to obtain, directly at the hospital, the final 3D models made by the engineers. This platform, developed and approved by European regulations, allows easy interaction between physicians and engineers for the creation of 3D models starting from the bi-dimensional preoperative images. Furthermore, the portal allows one to modify/personalize the model with presurgical information such as the planned resection plane or the clamping strategy; all of this information can be saved into the cloud system and be visualized later during the surgery. 

In this way, preoperative planning and intraoperative strategy are a continuum, increasing the quality of the surgical maneuvers, especially in a tailored setting. 

The possibility to have 2D and 3D images stored in a cloud platform can allow also the interaction between different specialists for remote teleconsultation. In fact, for example, the 3D model of a patient referred to a peripheral hospital can be to choose the best surgical strategy. 

This scenario can be included in the setting of telemedicine. This new kind of fruition of medical services has gained an ever-wider diffusion over the last years, especially thanks to the pandemic. The possibility to share medical contents using online platforms resulted to be a new useful instrument not only for patient and doctors’ consultation but also between physicians, increasing the equity of care among the different countries [24], especially in the training setting [25,26,27]. Lastly, we would like to focus on the object visualized into the ICON3D^TM^ platform: the 3D models. The models should respect high standards in terms of quality and reliability, and the process of 3D model reconstruction should follow rigorous steps in order to maximize the final product [11]. In fact, these models can be considered “medical devices”, therefore their high quality is mandatory, considering also that surgeons can change their surgical strategy on the basis of the information obtained. The models available today are the result of an evolution that has taken place over the last few years. If for prostate cancer the model was enriched with information regarding a tumour’s capsular contact, for kidney cancer the models are completely renewed with information regarding parenchymal vascular perfusion [18]. 

These refined 3D models allow surgeons to carry out a more accurate selective clamping, not based on the hypothetical arteries supplying the tumor, but driven by a mathematical demonstration of the perfusion areas. This implies also a change of perspective: in fact, to obtain an optimal selective clamping, we must not consider an artery’s direction towards the tumor, but the area of the tumor’s growth and the arteries supplying it. 

Some limitations of our work should be mentioned: firstly, our findings are the results of a self-assessed questionnaire, and the surgeons can be influenced by the specific event in which they have tried the technology; furthermore the responders were divided heterogeneously among the three groups with only 6 in the senior one. Clinical validation of this proposed technology is warranted in the next months. 

In the future, a larger adoption of 3D models’ guidance is desirable, encouraging the rise of preliminary clinical evidence [6,28]. A game changing role will be played by the adoption of artificial intelligence, starting from the 3D models’ creation [11] until the intraoperative setting with augmented reality surgery [29]. The possibility to have dedicated platforms such as MyMedics portal and ICON3D^TM^ rack at a surgeon’s disposal can facilitate the fruition of the 3D models easily with a postivie perception by the surgeons, as emerged in our findings.

## 5. Conclusions

The ICON3D^TM^ platform turned out to be a new tool for the fruition of 3D models both for the preoperative and intraoperative settings. Its ductility favors its use in both robotic and laparoscopic surgery, with a good perception by the surgeons as emerged with Health-ITUES and UEQ questionnaires.

## Figures and Tables

**Figure 1 medicina-59-00624-f001:**
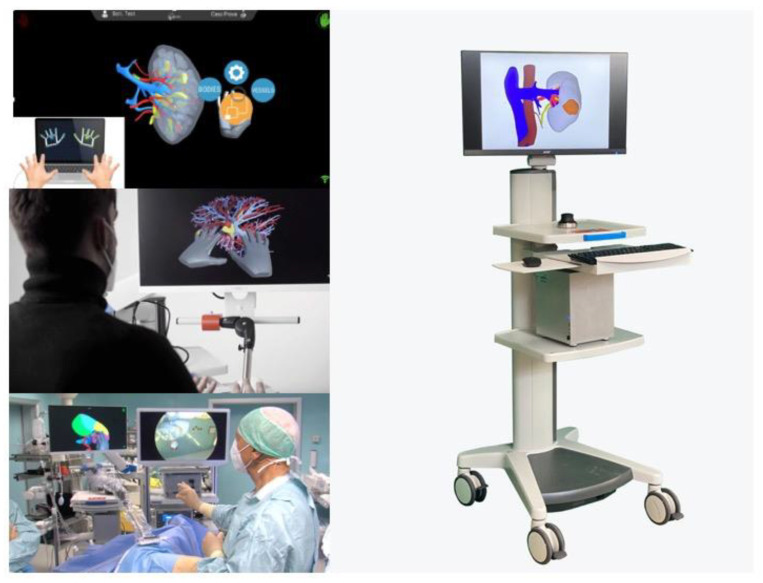
ICON3D^TM^ platform allows to visualize the 3D models and to interact with it in a touchless manner, even intraoperatively.

**Figure 2 medicina-59-00624-f002:**
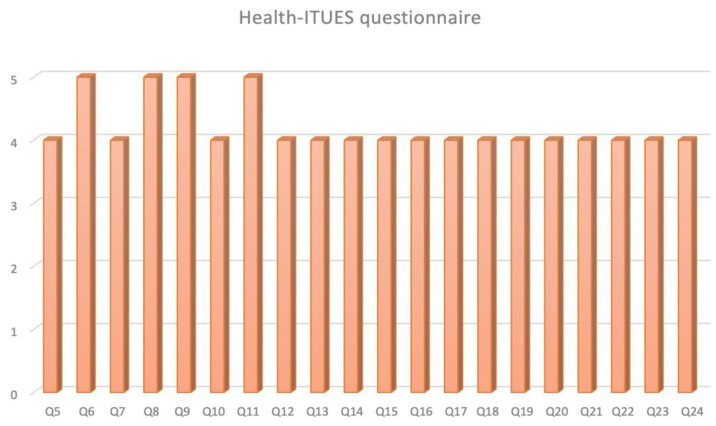
Overall results of Health-ITUES questionnaire.

**Figure 3 medicina-59-00624-f003:**
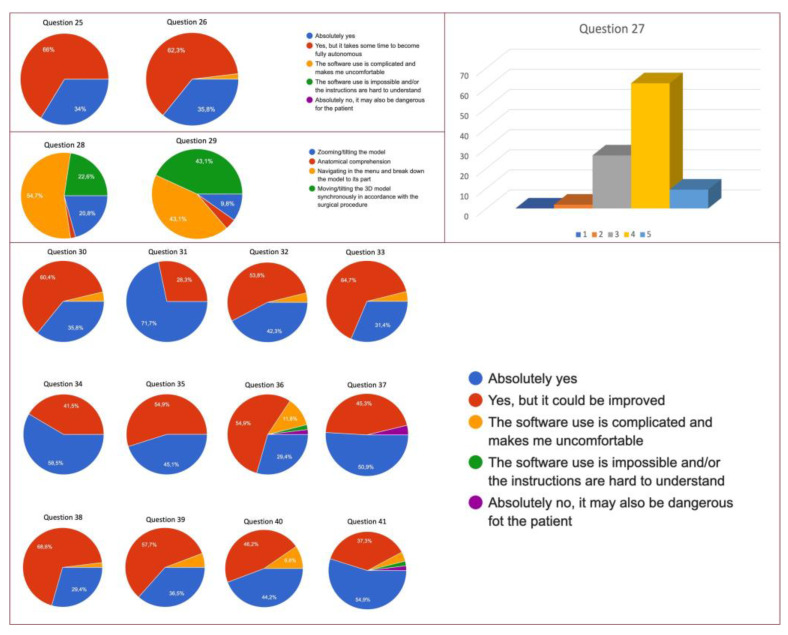
Overall results of UEQ questionnaire.

**Figure 4 medicina-59-00624-f004:**
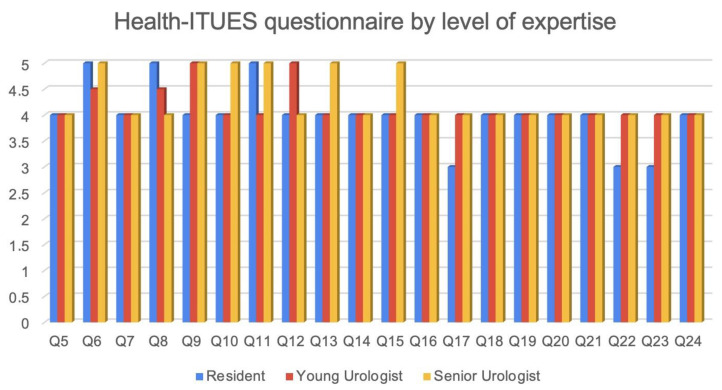
Results of Health-ITUES questionnaire stratified by level of expertise.

**Figure 5 medicina-59-00624-f005:**
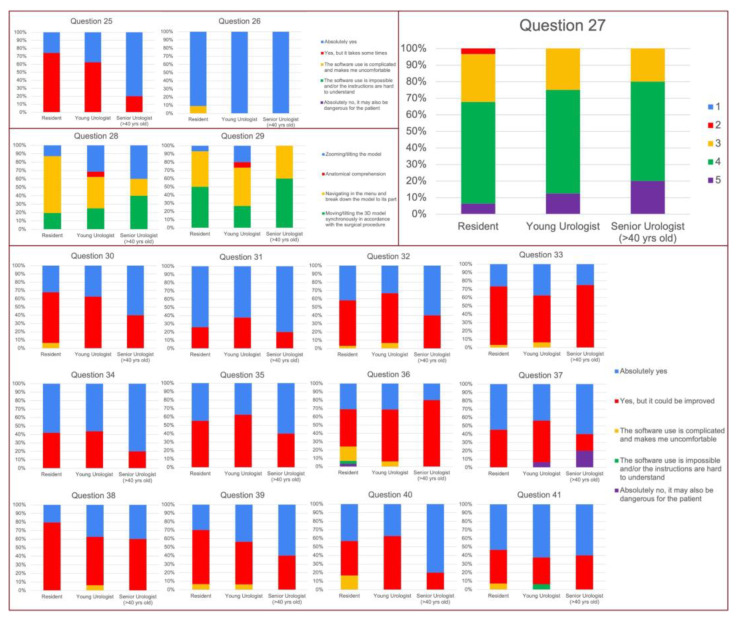
Results of UEQ questionnaire stratified by level of expertise.

**Table 1 medicina-59-00624-t001:** Participants’ demographics, level, and field of expertise.

Variable	Overall Participants
	(*n* = 53)
Gender, *n* (%)	
Male	37 (69.2)
Female	16 (30.8)
Level of expertise, *n* (%)	
Residents	31 (58.5)
Young urologist	16 (30.2)
Senior urologist	6 (11.3)
Surgical field of expertise, *n* (%)	
Open surgery	21 (42.9)
Laparoscopic surgery	21 (42.9)
Robotic surgery	38 (77.6)

## Data Availability

Data available on request due to restrictions e.g., privacy or ethical.

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
