# Peer review of "Health Information Technology Usability Evaluation Scale (Health-ITUES) and User-Experience Questionnaire (UEQ) for 3D Intraoperative Cognitive Navigation (ICON3DTM) System for Urological Procedures"

_medicina, 2023, doi:10.3390/medicina59030624_

Round 1

Reviewer 1 Report

This is a small-sample questionnaire study validating utility of novel pre- and intraoperative visualisation tool depicting the expected surgical field. I assume it is a first step of validating the tool since no clinically relevant endpoints (complications, margins, time of surgery, learning curve) were tested in the study. Thus the clinical value of the manuscript is rather modern.

The methodology is acceptable and the presentation is clear. Language is proper. Having in mind Journal scopus and short publishing history I feel the study can be addressed to Medicina even considering aforementioned limitations. 

Author Response

REVIEWER 1:

This is a small-sample questionnaire study validating utility of novel pre- and intraoperative visualisation tool depicting the expected surgical field. I assume it is a first step of validating the tool since no clinically relevant endpoints (complications, margins, time of surgery, learning curve) were tested in the study. Thus the clinical value of the manuscript is rather modern.

We thank the reviewer for this comment. We agree that our presented experience aimed to present for the first time this type of technology evaluating the surgeons’ perceptions. The real clinical validation will be object of future studies.

The methodology is acceptable and the presentation is clear. Language is proper. Having in mind Journal scopus and short publishing history I feel the study can be addressed to Medicina even considering aforementioned limitations.

 We thank the reviewer for this kind comment.

Reviewer 2 Report

A well-designed original study that aims to prove that the ICON3DTM platform is useful and accessible to guide surgeons in case planning both preoperatively and intraoperatively. The impression is that this tool has the potential to improve the surgeon's perception as well as the final surgical result. The article is well written and the analyzes well reported.

Conclusions are consistent with the evidence presented and addressed the main question posed.

Figures are ok.

Line 59: Correct ad → and

Congratulations for the work in this paper.

Author Response

REVIEWER 2

A well-designed original study that aims to prove that the ICON3DTM platform is useful and accessible to guide surgeons in case planning both preoperatively and intraoperatively. The impression is that this tool has the potential to improve the surgeon's perception as well as the final surgical result. The article is well written and the analyzes well reported.

 We thank the reviewer for this kind comment.

Conclusions are consistent with the evidence presented and addressed the main question posed.

 We thank the reviewer for this kind comment. 

Figures are ok.

 We thank the reviewer for this kind comment.

 Line 59: Correct ad → and

We correct this line according to your suggestion

Congratulations for the work in this paper.

We thank the reviewer for this kind comment.

Reviewer 3 Report

I congratulate the authors for this interesting manuscript.

There are some points I suggest clarifying.

In general check English

Please replace the verb enjoyed (in any form) when refer to the ICON3DTM in the whole text.

Add the limitations. For example, there only 6 participants in the senior group.

Abstract

Line 29 it seems that the word “evaluating” is misplaced

Introduction

Line 57 check grammar: the overlapped images can allows

Line 59: by ad, do you mean and?

Line 61 check grammar:  Focusing on prostate cancer the 3D models,

Material and methods

Use abbreviation of “residents, junior urologists, senior urologists” Re, etc

Surgical skills was a subjective?

Add that the participants completed the questionnaires to M and M.

Add the reference of the questionnaires. Are they validated for this task? Explain why did you choose them?

2.2 section

This section is very long and a big part is not relevant for the aim. Please shorten it.

Add reference to In fact, a surgeon 103 must follow a learning curve that takes time to walk in order to undertake an effective 104 "building in mind" process.

Results

Instead of using IQR, I suggest range or SD. IQR does not give a real idea of dispersion.

IN Evaluation of the different levels of expertise

When it comes to figure 4, the values described in the text are not in the figure (or laparoscopic skills (37.9% vs 35.7 vs 83% for residents, young urologists and senior urologists respectively, p=0.10) (Figure 4).   

Discusion

Please reduce redundant information.

Author Response

REVIEWER 3

I congratulate the authors for this interesting manuscript. 

There are some points I suggest clarifying. 

In general check English

We thank the reviewer for this comment. We have already revised the manuscript with the native speaker of our university; after your comment we did a second check

Please replace the verb enjoyed (in any form) when refer to the ICON3DTM in the whole text.

We thank the reviewer for this comment and change the manuscript accordingly.

Add the limitations. For example, there only 6 participants in the senior group.

We thank the reviewer for this comment and add this section at the endo of the discussion.

Abstract

Line 29 it seems that the word “evaluating” is misplaced

We thank the reviewer for this comment and correct the text

Introduction

Line 57 check grammar: the overlapped images can allows

We thank the reviewer for this comment and correct the text

Line 59: by ad, do you mean and?

We thank the reviewer for this comment and correct the text

Line 61 check grammar:  Focusing on prostate cancer the 3D models,

We thank the reviewer for this comment and correct the text

Material and methods

Use abbreviation of “residents, junior urologists, senior urologists” Re, etc

We thank the reviewer for this comment, and change the text accordingly

Surgical skills was a subjective?

We thank the reviewer for this comment. The main surgical skill was self-assessed by the responders.

Add that the participants completed the questionnaires to M and M.

We thank the reviewer for this comment and add this concept into the text

Add the reference of the questionnaires. Are they validated for this task? Explain why did you choose them?

We thank the reviewer for this comment. The questionnaire was already validated for this tasks and both were already mentioned in the text with Ref n° 12 and n°15.

2.2 section

This section is very long and a big part is not relevant for the aim. Please shorten it.

We thank the reviewer for this comment. Aimed to respect the journal guidelines for authors we opted to explain in detail the 3D model reconstruction process. Furthermore, we think that it is a key element and can be useful for the readers in order to understand how the models were done and how much the models are precise and how there are a real added value for the surgeons thanks to their high-definition quality.

Add reference to In fact, a surgeon 103 must follow a learning curve that takes time to walk in order to undertake an effective 104 "building in mind" process.

 We thank the reviewer for this comment and we add the Ref

Results

Instead of using IQR, I suggest range or SD. IQR does not give a real idea of dispersion.

We thank the reviewer for this comment, however we believe It might be better to present this categorical variable by median and IQR according to Dekking et al who stated that: “In descriptive statistics, the interquartile range (IQR) is a measure of statistical dispersion, which is the spread of the data”. (Ref: “Dekking, Frederik Michel; Kraaikamp, Cornelis; Lopuhaä, Hen Paul; Meester, Ludolf Erwin (2005). A Modern Introduction to Probability and Statistics. Springer Texts in Statistics. London: Springer London. doi:10.1007/1-84628-168-7. ISBN 978-1-85233-896-1.)

IN Evaluation of the different levels of expertise

When it comes to figure 4, the values described in the text are not in the figure (or laparoscopic skills (37.9% vs 35.7 vs 83% for residents, young urologists and senior urologists respectively, p=0.10) (Figure 4).   

 We thank the reviewer for this comment. We correct the place where we cited the Figure 4.

Discusion

Please reduce redundant information.

We try to reshape some part of the discussion reducing the redundancy with the results.

Round 2

Reviewer 3 Report

You have improved the manuscript a lot. 

It is not clear where for me how many atendess were the "99% of the attendees". 

Please avoid using the word enjoyed. such as in "had the possibility of enjoying the ICON3DTM experience"

Line 259, please delete resindence (Re), etc as these were already introduced before in the text

Author Response

You have improved the manuscript a lot. 

We thank the reviewer for this comment

It is not clear where for me how many atendess were the "99% of the attendees". 

We thank the reviewer for this comment. 99% of 53 partecipants. We changed this line aimed to clarify this point.

Please avoid using the word enjoyed. such as in "had the possibility of enjoying the ICON3DTM experience"

We thank the reviewer for this comment. We change the text accordingly.

Line 259, please delete resindence (Re), etc as these were already introduced before in the text

We thank the reviewer for this comment. We change the text accordingly.